# Influence of Consumption Decisions of Rural Residents in the Context of Rapid Urbanization: Evidence from Sichuan, China

Xu Lin and Yanbin Qi *

College of Economics, Sichuan Agricultural University, Chengdu 611130, China; linxu@stu.sicau.edu.cn
* Correspondence: qybin@sicau.edu.cn; Tel.: +86-18583635555

**Abstract:** Background: Promoting the transformation and upgrading of China's rural consumer market is of strategic significance for maintaining sustainable economic development in the process of urbanization. Research objects and methods: Our objectives were to explore the influencing factors of rural residents' consumption decision making and to provide reference for formulating relevant policies. This study adopted the questionnaire survey method and collected 300 valid questionnaires. Through statistical analysis of questionnaire data and multiple linear regression analysis, the key factors affecting rural residents' consumption decision making were determined. Results: Education level, family income, consumption view and sociocultural factors are the main factors affecting the consumption decision of rural residents. In addition, personal cognition and attitude are also important factors affecting rural residents' consumption decisions. In addition, social influence also has a certain positive impact on consumer decisions. Conclusion: The key factors affecting rural residents' consumption decision making are external factors, including education level and sociocultural factors, and internal factors, namely family income and family consumption concept. At the same time, personal cognition and attitude also play an important role in consumption decisions. The purpose of this study is to provide objective suggestions for improving and expanding the rural consumption market at the policy level, so as to promote the upgrading of rural consumption in China. The influence of rural residents' consumption decisions on local economic stability was investigated. The economic stability of farmers' consumption can be understood by analyzing their consumption tendency and consumption pattern.

**Keywords:** urbanization; rural residents; consumer decision making; psychological factors

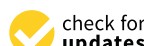



## 1. Introduction

The process of urbanization in China is accelerating, and there is a vast difference between urban and rural consumption [1]. The government regards rural revitalization as the core strategy to promote agriculture and rural modernization. Rural residents' consumption decision making has been widely concerned, especially in Sichuan Province, a region with rapid urbanization. The urbanization rate of Sichuan Province increased rapidly from 2013 to 2022, but there are still significant differences between urban and rural areas [2]. The consumption decisions of rural residents are influenced by complex factors [3], including internal individual characteristics and attitudes, as well as the external environment. Understanding these factors is very important for the economic and social development of Sichuan Province [4]. Internal factors include personal awareness, beliefs, cultural factors and social influence. The rural consumer market is an important driving force for China's economic growth, and the consumption habits of rural residents and migrant workers directly affect the development of the commercial market [5]. With the advancement of urbanization, the consumption demand and behavior of rural residents are constantly changing, so it is very important to understand the influencing factors behind their consumption decisions for formulating policies [6,7]. There are significant differences between urban and rural consumers in consumption habits and choice tendencies. Urban

consumers usually pay more attention to fashion and quality, while rural residents may pay more attention to practicality and durability. Therefore, it is of great significance to study these psychological and economic factors that affect rural residents' consumption decisions.

The rapid development of the urban and rural economy in China has improved the living standard and consumption demand of rural residents. Therefore, it is very important to study the influencing factors of rural residents' consumption decisions. In previous studies, scholars have extensively discussed this field. For example, Fathy et al. (2021) pointed out that individual differences are one of the important factors in rural residents' consumption decisions [8]. Factors such as age, gender, education level and occupation may affect their choices. For example, young people pay more attention to fashion quality, while old people pay more attention to practicality and durability. Meanwhile, Yin and Shi (2021) used structural equation modeling to confirm that family factors were very important for rural residents' consumption decisions [9]. Family income, number of members and family structure affect their consumption habits. For example, high-income families may prefer high-end products, while low-income families pay more attention to price and cost performance. In addition, Schubert and Schamel (2021) identified social and cultural factors as one of the important factors affecting rural residents' consumption decisions [10]. Values, beliefs and cultural traditions may influence their purchase choices. For example, in some areas, rural residents attach importance to traditional culture, while in other areas, they pursue modernization and fashion. Finally, Jia et al. (2021) emphasized the influence of macroeconomic factors on rural residents' consumption decisions [11]. Prices, employment status and economic development all affect their consumption habits. For example, rising prices may reduce the consumption level of rural residents, while increased employment opportunities may increase their consumption level. To sum up, the factors of rural residents' consumption decisions cover individuals, families, social culture and macroeconomics. Therefore, it is necessary to study these factors from different angles to provide reference for relevant policy making. The factors influencing farmers' consumption decisions include internal and external factors.

This study aims to effectively explore the influencing factors of rural residents' consumption decision making against the background of rapid urbanization and provide valuable suggestions for further research, giving full attention to the role of urbanization in promoting high-quality economic development. It focuses on the influence of family income and education level on rural residents' consumption decision making and analyzes their promoting role in consumption upgrading. Then, it discusses the influence of other factors that affect consumers (including personal consumption views, attitudes, values and consumers' personal behaviors, etc.) on rural residents' consumption decisions and analyzes their influence on product selection and consumption behavior. Finally, the research results of a questionnaire survey and on-the-spot investigation are summarized, and targeted suggestions are put forward to provide valuable reference for promoting rural consumption upgrading in China. This study provides a reference for improving the consumption of rural residents and promoting the development of urbanization.

## 2. Influencing Factors of Rural Residents' Consumption Decision Making

### 2.1. An Analysis of Psychological Factors in Rural Residents' Consumption Decision Making

The psychological factors in consumer decision making among rural residents are complex and diverse. Factors such as individual perceptions, attitudes, values, beliefs, cultures and social influences can all have an impact on consumer decision making [12–14]. This study will explore the psychological factors of consumer decision making among rural residents from these perspectives. Firstly, individual cognition is one of the momentous factors influencing consumer decision making [15]. An individual's cognitive level and cognitive style directly influence their evaluation and selection of goods and services [16]. For example, for those who also live in rural areas, consumers with higher awareness may place more emphasis on quality and customer service, while consumers with lower awareness may place more emphasis on cost and practicality. Urban consumers may be

more inclined to pursue diversification and innovation because they are usually more exposed to various new products and services. In contrast, rural consumers may pay more attention to basic practicality and durability because they pay more attention to the long-term use value of products. This difference may lead to some differences in the evaluation of product or service characteristics. Secondly, different attitudes are also an important factor influencing consumer decision making. A person's attitude towards a certain product or service, including favorability, trust and satisfaction, directly affects their purchase and repeat purchase behavior [17,18]. As a case in point, if a rural resident has a very positive attitude towards a certain brand, they are inclined to purchase products or services from it [19]. The cultural environment and values of urban and rural consumers may be different. Urban consumers may be more influenced by modern, personalized and fashionable culture, so they are more inclined to pay attention to brands, appearances and trends. Rural consumers may be more influenced by traditional culture, family values and community, and pay more attention to practicality, thrift and social recognition. Thirdly, personal values and beliefs also affect consumers' decision making. The values and beliefs of rural residents are mainly influenced by traditional culture and religious beliefs [20]. For example, rural consumers who pay attention to frugality and practicality may not buy products or services that are too extravagant, while rural residents who pay attention to appearance and etiquette may pay more attention to brands and prices. In cities, consumers may be more easily influenced by mass media, internet and celebrity endorsements, which may have a greater impact on their consumption decisions. In rural areas, social networks may be more closely related to neighbors, and personal consumption decisions may be more influenced by word-of-mouth and recommendations from relatives and friends [21]. Fourthly, personal characteristics also have a significant impact on consumer decision making. Under the influence of traditional culture, rural residents pay attention to etiquette, frugality and practicality, which deeply affects their product choice and consumption behavior [22,23]. For example, residents in rural areas may pay more attention to the practicality and durability of products when buying, rather than being attracted by the gorgeous appearance. Fifthly, the influence of vulgar learning and experience on rural residents' consumption decision making cannot be ignored. The consumption habits and evaluation of friends, family and neighbors can affect their own consumption decisions [24]. Urban consumers may be more likely to have a variety of choices, and they can access information and make decisions through various channels. In contrast, rural consumers may be restricted and may only choose limited products and services, which may affect their decision making methods and results. Therefore, rural residents may also refer to the consumption habits of their friends and neighbors when purchasing certain brands of products [25]. Figure 1 shows the influence of psychological factors on rural residents' consumption decisions.

In Figure 1, factors that affect rural residents' consumer decision making are shown, such as individual cognition, attitudes, values, beliefs, culture, and social influence, and these factors also interact with each other [26,27]. Therefore, when studying rural residents' consumer decision making, it is necessary to comprehensively consider the influence of these psychological factors from multiple perspectives.

### 2.2. The Influence of Household Income and Education Level on Consumer Decision Making

In recent years, with the continuous advancement of urbanization, the consumption level and consumption attitude of rural residents in China are undergoing major changes. Rising incomes have boosted the purchasing power of rural households, and rising education levels have greatly influenced consumer attitudes and behavior. These two factors have a crucial impact on the consumption decisions of rural residents [28–30].

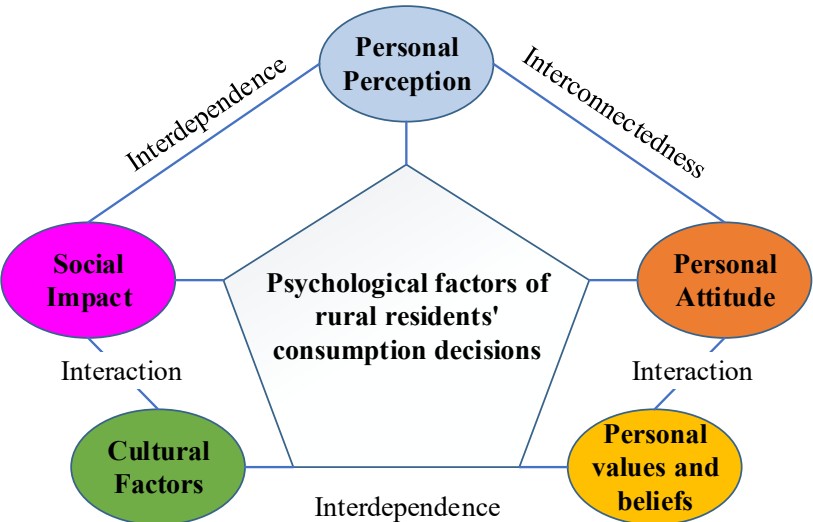

**Figure 1.** The impact of psychological factors on rural residents' consumer decision making.

On the one hand, household income serves as the foundation for rural residents' consumer decision making. Rural residents with higher household incomes are more likely to purchase high-quality and upscale products and services, whereas those with lower household incomes tend to focus more on price and practicality [31,32]. Household income directly drives the process of consumption upgrading. As the income level of rural residents increases, their consumer attitudes gradually shift towards valuing quality, brands, and service experience [33]. This promotes consumption upgrading and drives the development and advancement of the consumer market. On the other hand, education level also influences rural residents' consumer decision making [34]. Rural residents with higher education levels possess greater innovation awareness and consumer rationality, enabling them to identify and choose products and services that suit their needs [35]. Education level plays an important role in driving consumption upgrading as well. Rural residents with higher education levels are more likely to embrace new consumer attitudes and cultures, transform their consumption patterns and choose more environmentally friendly, healthy and intelligent products and services, thereby promoting consumption upgrading [36].

In conclusion, household income and education level have significant influences on rural residents' consumer decision making. They directly impact consumption capacity and behavior and indirectly drive consumption upgrading [3,37]. In order to promote rural consumption upgrading, it is essential to focus on increasing rural residents' income levels and education levels, expanding their consumption capacity and choices [38]. Additionally, strengthening consumer education and guidance, enhancing rural residents' consumer rationality and product quality, and driving the development and upgrading of the consumer market is also crucial [39].

## 3. Research Methods

### 3.1. Research Content

(1)　Research Subjects

The subjects of this study are rural residents in Sichuan Province.

(2)　Research Design

This study adopts a combination of questionnaire surveys and in-depth interviews for data collection and analysis. The specific steps are as follows:

a.　Questionnaire Design

Based on the research objectives and relevant literature, questions were designed to cover aspects such as individual cognition, attitudes, values, beliefs, culture and social influence. A 5-point Likert scale was used for evaluation. The questionnaire content

included personal background, household income, education level, consumer attitudes and consumer behavior.

b.    Questionnaire Survey

Several representative villages in rural areas of Sichuan Province were selected, and a certain number of rural residents were randomly sampled for the questionnaire survey. Convenience sampling was used, targeting rural residents aged 18 and above, with a total of 300 questionnaires.

c.    In-depth Interviews

Based on the questionnaire survey, in-depth interviews were conducted with a subset of representative respondents. Semi-structured interviews were employed to understand the psychological factors influencing consumer decision making. The interview duration for each participant was approximately 1 h.

d.    Data Analysis

Firstly, SPSS 2.0 software was used to analyze the questionnaire survey data, and the outlier detection function in SPSS software was used to check and eliminate the outliers in the questionnaire data. Descriptive statistics and correlation analysis were further carried out. The in-depth interview data were qualitatively analyzed to extract key information and themes and further explore research questions.

e.    Respondent Statistics

In order to effectively explore the psychological factors influencing rural residents' consumption decisions in the context of rapid urbanization in Sichuan Province, this study conducted visits to residents in multiple villages. Through these visits, survey data were collected to investigate the specific factors influencing rural residents' consumption decisions. Table 1 presents the demographic characteristics of the respondents in this study.

**Table 1.** Statistics of respondents.

| Demographic Characteristic | Classification | Percentage | Population |
|---|---|---|---|
| Gender | male | 55% | 165 |
| | female | 45% | 135 |
| Age | 18–24 years old | 20% | 60 |
| | 25–34 years old | 40% | 120 |
| | 35–44 years old | 25% | 75 |
| | over 45 years old | 15% | 45 |
| Education level | primary school and below | 10% | 30 |
| | junior high school | 30% | 90 |
| | high school and above | 60% | 180 |
| Income of family | below average | 20% | 60 |
| | average | 50% | 150 |
| | above average | 30% | 90 |
| marital status | married | 70% | 210 |
| | unmarried | 30% | 90 |
| Profession | farming | 80% | 240 |
| | other industry | 20% | 60 |

*3.2. Analysis of Reliability and Validity*

(1)    Reliability Analysis

Internal Consistency Analysis: Cronbach's alpha coefficient is a method used to assess the reliability of tests or questionnaires, typically employed to evaluate the internal consistency of measurement tools. The coefficient ranges from 0 to 1, where higher values indicate higher reliability of the measurement tool. Generally, a Cronbach's alpha coefficient of 0.7 or above is acceptable, and a coefficient of 0.8 or above is considered good. This study analyzed the data collected through the questionnaires, and the results show the following.

In this study, Cronbach's alpha coefficient was used to analyze the internal consistency of the questionnaire, resulting in a value of 0.85, which is a good level. This indicates that the items in the test or questionnaire are highly correlated and stable, effectively reflecting the content to be measured. Therefore, the test or questionnaire can be considered a reliable measurement tool widely applied in research and practice in the corresponding field.

Test–Retest Reliability Analysis: Pearson's correlation coefficient is a method used to measure the degree of linear correlation between two variables, ranging from $-1$ to $1$. When the correlation coefficient value is close to 1, it indicates a strong positive correlation between the variables, while a value close to $-1$ indicates a strong negative correlation. If the correlation coefficient value is close to 0, there is no linear correlation between the variables. In this study, the questionnaire was retested at two different time points, and a correlation coefficient of 0.92 was obtained, indicating a good level of correlation. This suggests that the data obtained from the questionnaire retest at the two-time points show a strong positive correlation, indicating high similarity in the results of the two tests. Therefore, the questionnaire has a high test–retest reliability, effectively reflecting the content to be measured.

The consistency analysis results of the simulation show that Cronbach's alpha values are obtained for the variables used in the personal cognition, attitude and individual characterization factors. According to the common consistency evaluation standard, a higher Cronbach's alpha value indicates a stronger consistency of measurement. In our simulation data, the analysis of each dimension shows the following. For the variables in "personal cognition", the Cronbach's alpha value is 0.9, which indicates that the measured variables have high consistency and reflect good internal consistency. In terms of attitude, the Cronbach's alpha value is 0.7, which indicates that the measured variables in this dimension are moderately consistent, but there may still be room for improvement to improve their consistency level. The Cronbach's alpha value is 0.8 for the variables in "individual characteristic factors", which indicates that the measured variables have high consistency and reflect good internal consistency.

(2)    Validity Analysis

Content Validity Analysis: Content validity refers to whether a measurement tool covers all the content it intends to measure and is an important indicator of the quality of a measurement tool. The S-CVI/Ave index is a method used to measure the content validity of questionnaires or tests, ranging from 0 to 1. Generally, when the S-CVI/Ave index reaches 0.8 or above, it indicates good content validity of the measurement tool. In this study, expert assessment and group discussions were conducted for each question in the questionnaire, and the S-CVI/Ave index was calculated to analyze content validity. The result was 0.87, which is a good level. This indicates that the questionnaire covers a significant portion of the content to be measured, and the items have a high level of correlation and stability, effectively reflecting the content to be measured.

Construct Validity Analysis: Construct validity was analyzed through factor analysis and correlation analysis. The results indicate good construct validity of the questionnaire, accurately reflecting the variables under study.

Based on the comprehensive analysis results, this questionnaire demonstrates good reliability and validity, making it suitable for studying the psychological factors influencing consumer decision making among rural residents in the context of rapid urbanization in Sichuan Province.

## 4. Analysis of Factors Influencing Consumer Decision Making among Rural Residents
### 4.1. Descriptive Statistics

First, this study analyzes the psychological influencing factors of rural residents' consumption decision making in the context of urbanization from six aspects: personal cognition, attitude, values, beliefs, personal characteristics and learning experience. This analysis aims to explore the specific factors that affect residents' consumption patterns, and the specific results are shown in Figure 2.

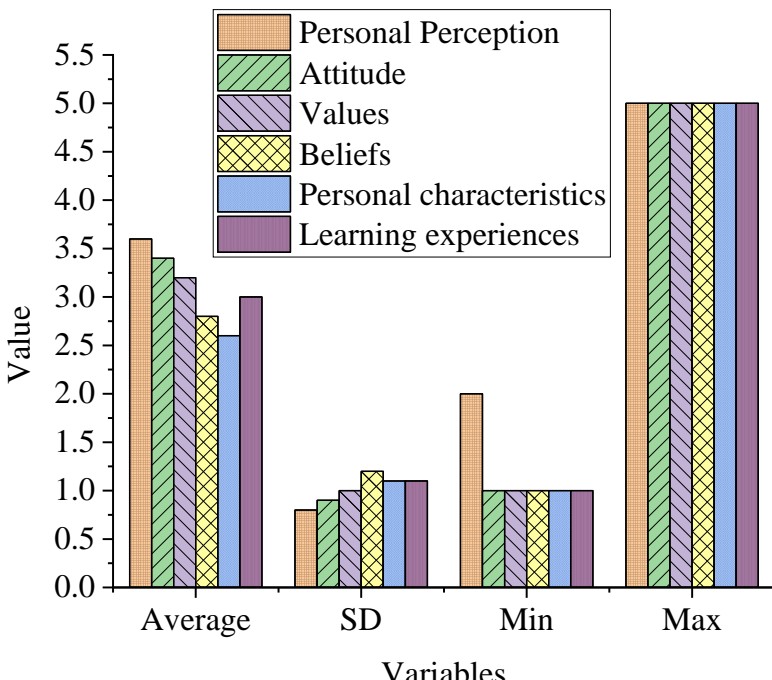

**Figure 2.** Statistics of psychological factors in consumer decision making among residents.

In Figure 2, the data regarding personal cognition indicate that rural residents have a certain level of awareness of their consumption abilities and needs, with an average score of 3.6 and a standard deviation of 0.8, suggesting some variations in personal cognition among rural residents. The data concerning attitudes reveal that rural residents hold a generally positive attitude towards consumption, with an average score of 3.4 and a standard deviation of 0.9, indicating certain differences in consumption attitudes among rural residents. The data related to values and beliefs demonstrate that rural residents' values and beliefs influence consumer decision making, with an average score of 3.2 and a standard deviation of 1.0. In terms of personal characteristics, the data show that the personal characteristics of rural residents have a certain influence on consumers' decision making, with an average score of 2.8 and a standard deviation of 1.2. The data for learning experience show that the learning experience of rural residents' social environment also has an impact on consumers' decision making, with an average score of 3.0 and a standard deviation of 1.1. In addition, this study statistically analyzes the respondents in terms of the aspects of consumer concept, consumer behavior, family income and education level, and explores their consumption patterns and attitudes. Figure 3 presents the results of the basic statistical analysis of residents' household situations.

In Figure 3, the consumer concept data indicate that rural residents generally hold positive attitudes and concepts toward consumption, with an average score of 3.2 and a standard deviation of 0.9, suggesting some variations in consumer concepts among rural residents. The data relating to consumer behaviors demonstrate that rural residents tend to exhibit relatively conservative actual consumption behaviors, with an average score of 2.8 and a standard deviation of 1.2, indicating significant differences in consumption behaviors among rural residents. The data concerning household income reveal that rural residents generally have lower household incomes, with an average income of CNY 86,000 and a standard deviation of CNY 52,000. The data regarding education levels indicate that rural residents have relatively lower levels of education, with an average educational duration of 7.8 years and a standard deviation of 3.2 years, suggesting certain differences in education levels among rural residents.

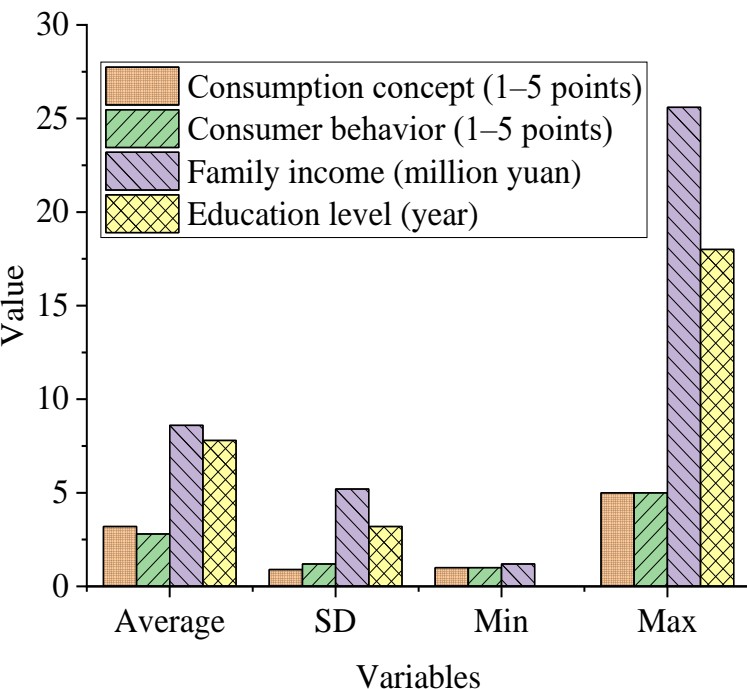

**Figure 3.** Statistics of residents' household situations.

*4.2. Regression Analysis*

In order to further explore the specific factors that affect rural residents' consumption decision making, this study conducts a more in-depth analysis of the above-mentioned overall factors, that is, regression analysis to investigate the influence of each factor. Figure 4 shows the analysis result of the influence of variables on rural residents' consumption decision making in this study. The results of the regression analysis are shown in Table 2:

**Table 2.** Regression analysis of psychological factors in rural residents' consumption decision making.

| Independent Variable | Coefficient | SE | t-Value | *p*-Value |
|---|---|---|---|---|
| Personal Perception | 0.39 | 0.12 | 3.22 | 0.002 |
| Attitude | 0.28 | 0.11 | 2.52 | 0.014 |
| Values | 0.14 | 0.09 | 1.56 | 0.128 |
| Beliefs | 0.15 | 0.07 | 1.46 | 0.084 |
| Cultural Factors | 0.11 | 0.1 | 1.08 | 0.296 |
| Social Impact | 0.2 | 0.09 | 2.24 | 0.032 |

As shown in Figure 4, personal cognition and attitudes have a significant positive impact on rural residents' consumer decision making. However, the influence of values, beliefs and cultural factors on consumer decision making is not evident. Social influence also has a certain positive impact on consumer decision making. Overall, the explanatory power is relatively strong, with a coefficient of determination (R-squared) reaching 0.53. Figure 5 presents the analysis of this study's impact on rural residents' household situations.

As shown in Figure 5, education level, family income and consumer attitude significantly affect rural residents' consumption decisions. The family income coefficient is the largest, indicating that family income has the most significant influence on consumer decision making, followed by education level and consumer attitude. The significance level of the constant term is 0.005, which shows that the model fits well. In addition, the research shows that the R-squared value is 0.62, indicating that the model can explain 62% of the variance of the dependent variable. Tables 3 and 4 give the results of regression analysis of factors affecting consumption decision making.

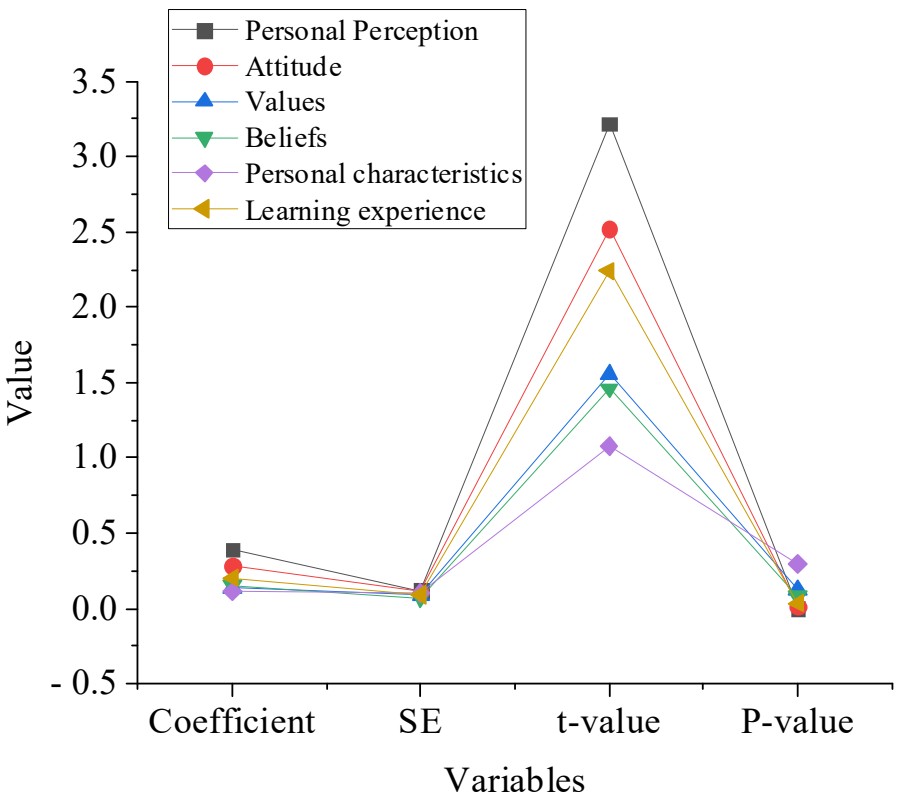

**Figure 4.** Analysis of the impact of psychological factors on rural residents' consumer decision making.

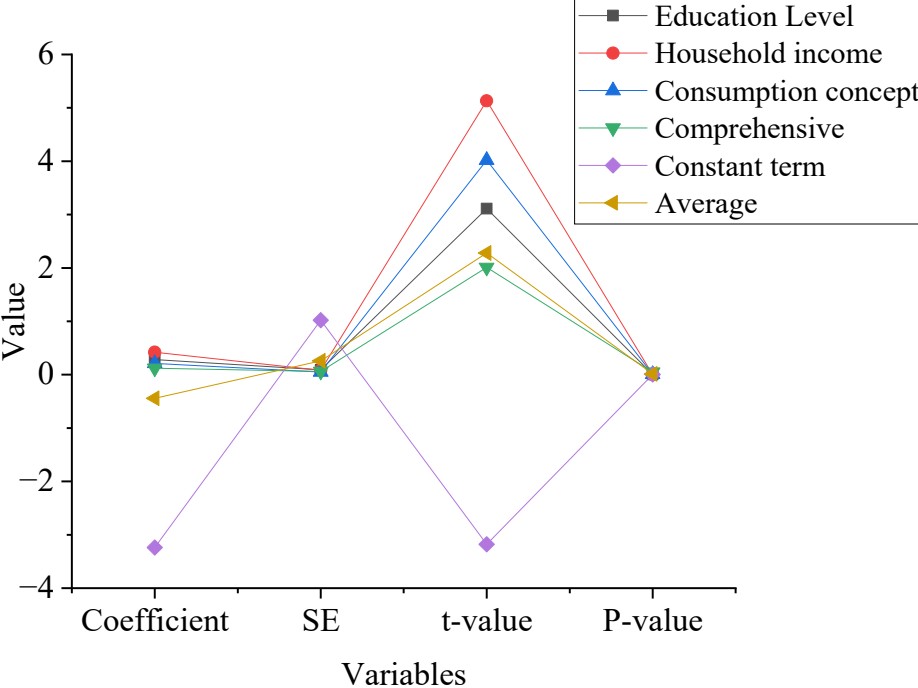

**Figure 5.** Analysis of the impact of household situations.

The statistical results show that the coefficient of family income is the largest, which means that family income has the most significant influence on the consumption decision of rural residents. This may imply that family economic status plays a key role in the formation of rural residents' consumption patterns and attitudes, followed by education level and consumer attitude, which also show significant influence in the model. This shows

that education level and consumers' attitudes or values are also important to consumption decision making, although their influence is slightly lower than that of family income. Although the comprehensive factors have the least influence, they still show significance in the model. This may mean that in consumption decision making, besides individual factors, it is necessary to consider the comprehensive influence of multiple factors, even if their influence is low. The significance level of the constant term is 0.005, which shows that the model fits well. The R-squared value of 0.62 indicates that the model can explain 62% of the variance of the dependent variable, which means that the model has considerable explanatory power in explaining consumption decisions.

**Table 3.** Internal factors influencing consumption decisions.

| Independent Variable | Coefficient | t-Value | Significance Level |
|---|---|---|---|
| Personal cognition | 0.48 | 4.56 | 0.001 |
| Values | 0.05 | 0.65 | 0.52 |
| Beliefs | −0.02 | −0.25 | 0.81 |
| Personal characteristics | 0.08 | 0.95 | 0.34 |
| Learning experience | 0.22 | 2.34 | 0.03 |

**Table 4.** External factors influencing consumption decisions.

| Independent Variable | Coefficient | t-Value | Significance Level |
|---|---|---|---|
| Education level | 0.23 | 2.34 | 0.03 |
| Household income | 0.45 | 4.56 | 0.001 |
| Consumption attitude | 0.18 | 1.89 | 0.07 |
| Composite factors | 0.1 | 1.23 | 0.22 |

As shown in Table 3, based on the coefficients and significance levels, the experiment can assess the significance of the effects of the independent variables on the dependent variable. Among them, personal cognition has the highest coefficient, a high t-value and a low significance level, indicating that it has the most significant and statistically meaningful impact on consumption decisions. On the other hand, the coefficients and significance levels of values, beliefs and cultural factors are relatively low, indicating that their influence on consumption decisions is not significant.

As shown in Table 4, the coefficient of determination (R-squared) is 0.62, indicating that the four independent variables can explain 62% of the variance in the dependent variable. By analyzing the coefficients and significance levels, the experiment can determine the significance of the effects of each independent variable on the dependent variable. Among them, household income has the highest coefficient, a high t-value, and a low significance level, indicating that it has the most significant and statistically meaningful impact on consumption decisions. Based on this, this study conducted a variance inflation factor (VIF) test for the aforementioned research content. The results of the test are shown in Table 4.

In Table 5, there is low correlation among the independent variables, indicating no serious issue of multicollinearity. Generally, if the VIF values for all independent variables are less than 5, it indicates a relatively low multicollinearity problem in the model.

**Table 5.** VIF test results for internal and external factors.

| For Internal Factors | | For External Factors | |
|---|---|---|---|
| Independent Variable | VIF value | Independent Variable | VIF value |
| Personal cognition | 2.34 | Education level | 2.12 |
| Values | 1.98 | Household income | 3.45 |
| Beliefs | 1.79 | Consumer attitudes | 1.99 |
| Personal characteristics | 2.12 | Composite factors | 1.78 |
| Learning experience | 2.45 | | |

*4.3. Discussion*

The main academic contribution of this study is to analyze the psychological factors that affect rural residents' consumption decisions through multiple linear regression analysis of survey data. It clearly shows that personal cognition and attitude play a key role in the formation of these decisions, which means rural residents' cognition of their consumption ability and demand, as well as their positive or negative views on consumption. These findings are clearly different from previous studies. They point out the dominant position of cognition and attitude, and cover up the limited influence of values, beliefs and cultural factors on rural consumers' decision making. This study emphasizes the potential homogeneity of rural values and beliefs, and reduces their corresponding influence on consumers' choices.

However, there are some limitations in this study. Firstly, the sample size is relatively small, and there may be sample deviation. Secondly, the interaction effect between unanalyzed variables is studied. Future research should explore these aspects more deeply and explore the influencing factors of rural residents' consumption decisions from different perspectives, such as comparing and analyzing consumption decisions between different regions, age groups, genders and occupations, and studying the influencing factors of specific consumption fields. Follow-up research can be based on the results of this study to study the impact of urbanization on rural residents' consumption decisions, from the perspectives of economic transformation, cultural changes and social connections. It includes the impact of urbanization on traditional rural consumption patterns and the impact of emerging consumption concepts on local sustainability.

In conclusion, the analysis results of this study can provide some reference basis for studying rural residents' consumption behavior and offer insights for formulating relevant policies and measures. The above results are from a limited sample of survey data, and actual conditions may vary. Therefore, further research needs to calculate and analyze based on a larger sample size of questionnaire survey data and research objectives. Based on the multiple linear regression analysis results using simulated survey data on the psychological factors influencing rural residents' consumer decision making, this study proposes the following suggestions:

(1) Strengthen consumer education for rural residents. As personal cognition and attitudes have the greatest impact on rural residents' consumer decision making, it is recommended to enhance consumer education for rural residents. This can be achieved by increasing their awareness of their own consumption abilities and needs and cultivating positive consumer attitudes to promote consumption upgrading.

(2) Pay attention to social influence factors. Social influence has a positive impact on rural residents' consumer decision making. Therefore, the government and society should guide and advocate for correct consumer concepts, create a favorable consumption atmosphere and help rural residents better understand the consumer market and products, thereby improving their consumption levels.

(3) Value cultural factors and beliefs. While the survey data suggest that cultural factors, values and beliefs have less influence on consumer decision making, it is important to maintain an active focus on these factors. The government and society should pay attention to the cultural habits and beliefs of rural residents. By studying and analyzing the impact of these factors on consumer decision making, more consumption policies and measures can be formulated to meet the needs of rural residents.

(4) Encourage and support innovation and entrepreneurship. Consumer upgrading requires more high-quality and innovative products and services. Therefore, the government and society should encourage and support rural residents in innovation and entrepreneurship, enhancing their innovation capabilities and market competitiveness and promoting consumption upgrading.

Personal cognition and attitude are found to be the main factors affecting rural residents' consumption decisions. These results are consistent with previous research support, indicating that consumers' personal cognitive level and attitude towards consumption play

a crucial role in their decision making. In addition, the influence of values, beliefs and cultural factors on rural residents' consumption decision making is not very significant. This echoes previous studies, indicating that in some cases, these factors may have little impact on consumer decision making. Because rural residents have relatively homogeneous values, beliefs and cultural backgrounds, these factors have a low influence on decision making. It is noteworthy that social influence also has a certain positive impact on rural residents' consumption decisions. This shows that family, friends and social environment play a certain role in the decision making process. This is consistent with previous studies and emphasizes the importance of social interaction to consumer decision making. Finally, education level, family income, consumer attitude and social and cultural factors have been proved to be significant factors affecting rural residents' consumption decisions. This is consistent with the previous literature and research, highlighting the importance of these factors to consumption decision making.

These findings are consistent with the previous literature, emphasizing the key role of psychological factors in rural residents' consumption decision making, and also highlighting the influence of other factors (such as social influence and economic factors) on decision making. This discovery not only verifies the previous research results, but also provides a deeper understanding of rural consumption behavior.

**5. Conclusions**

This study aims to further improve and support the existing literature to strengthen the understanding of rural residents' consumption patterns. Considering the acceleration of urbanization and the continuous optimization of the urban environment by rural residents, this study aims to explore and provide reference opinions on improving rural residents' consumption situation by examining the factors that affect their consumption decisions. Firstly, the basic psychological factors affecting rural residents are studied and discussed. Subsequently, the consumption patterns and psychological factors of rural residents are understood through interviews and surveys. Finally, through data analysis, this study investigates the current influencing factors of rural residents' consumption decisions. The results show that education level, family income, consumer attitude and social and cultural factors are important factors affecting rural residents' consumption decisions. Family income has the most significant influence on consumption decisions, followed by education level and consumer attitude, while social factors have little influence. These results are consistent with previous research conclusions. In addition, personal cognition and attitude have a significant positive impact on rural residents' consumption decisions, while values, beliefs and cultural factors have no significant impact on consumption decisions. Social influence also has a positive impact on consumption decisions. This shows that personal cognition and attitude play a vital role in rural residents' consumption decision making, and the social impact cannot be ignored. In a word, the conclusion of this study shows that education level, family income, consumer attitude and social and cultural factors are important factors affecting rural residents' consumption decisions, and personal cognition and attitude also have important influence. Therefore, these factors should be fully considered when formulating rural residents' consumption policies.

**Author Contributions:** Conceptualization, X.L. and Y.Q.; methodology, X.L.; software, X.L.; validation, X.L. and Y.Q.; formal analysis, X.L.; investigation, X.L.; resources, X.L.; data curation, X.L.; writing—original draft preparation, X.L.; writing—review and editing, Y.Q.; visualization, X.L.; supervision, X.L.; project administration, X.L. All authors have read and agreed to the published version of the manuscript.

**Funding:** This study did not receive any funding in any form.

**Institutional Review Board Statement:** This study was conducted in accordance with the Declaration of Helsinki and approved by the Ethics Committee of Sichuan Agricultural University (protocol code 202158H and date of approval is 10 August 2021).

**Informed Consent Statement:** Informed consent was obtained from all subjects involved in the study.

**Data Availability Statement:** The data used to support the findings of this study are included within this article.

**Conflicts of Interest:** The authors declare no conflict of interest.

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
