# Peer review of "Influence of Consumption Decisions of Rural Residents in the Context of Rapid Urbanization: Evidence from Sichuan, China"

_sustainability, doi:10.3390/su152316524_

Round 1
Reviewer 1 Report
Comments and Suggestions for Authors
Congratulations to the authors, the paper is very well conceived.
The conclusions can be further improved and justified with existing literature to reinforce their results.
Author Response
Reviewer 1
Comments and Suggestions for Authors
Congratulations to the authors, the paper is very well conceived.
The conclusions can be further improved and justified with existing literature to reinforce their results.
Reply: Thank you for your suggestions. According to your suggestions, we have updated the conclusion part to further analyze the existing literature and further improve the relationship between this paper and the ongoing research.
Reviewer 2 Report
Comments and Suggestions for Authors
Introduction and Motivation
· Somewhere in the abstract, the author’s say the following:
“Results: Education level, family income, consumption concept and social and cultural factors are the main factors that affect rural residents' consumption decision-making. Furthermore, Individual cognition and attitude are also important factors affecting rural residents' consumption decision-making, while values, beliefs, and cultural factors have less obvious effects on consumer decision-making.
There is a contradiction about the impact of cultural factors.
Also, the introduction is hard to follow: The author’s use phrases such as External factors, Internal factors, Phycological factors, Culture, Socio-economic status, Family income and education, etc. These phrases are used in great deal, in repeated fashion throughout the introduction, and seem to be very central to this research. However, they are used so disjointedly that I am unable to get a clear understanding of what motivates this research. Simply put, the ideas don’t flow naturally. I suggest that the author rewrites the introduction in a clear and concise language. In doing so, the authors should highlight, in a less confusing way, the so-call psychological factors and economic factors that are driving the consumption decisions in rural households. The authors can then explain how that is different for urban consumers.
Literature Review
· The authors cite some literature that focus on household consumption decisions in the rural areas. Some information on how that contrasts with urban consumers would be interesting to see.
Section 2:
· Section 2 is titled “Psychological Factors of Consumer Decision-Making Among Rural Residents”. However, there is a sub-heading (2.1) on phycological factors and another sub-heading (2.2) on household income and education level. Is it the author’s proposition that household income and education level are part of psychological factors? If so, my concerns about the language in the introduction is even larger. If not, the authors should rewrite the heading in section 2 to reflect both factors.
Research Methodology
Consider using bullet points, roman numerals, or alphabets for the list of steps under “Research Design”. The current use of numbers makes it confusing to follow.
Analysis
The analytical section (section 4) is much clearer. The discussion is easier to follow, particularly with the help of the Tables and Figures. Also, Tables 2-4 make clear distinction between internal and external factors influencing consumption decisions. Fig. 2 and Fig. 4 also put more light on the so-called psychological factors. I would like to see this clarity reflected in the introductory part of the paper.
Author Response
Reviewer 2
Comments and Suggestions for Authors
Introduction and Motivation
- Somewhere in the abstract, the author’s say the following:
“Results: Education level, family income, consumption concept and social and cultural factors are the main factors that affect rural residents' consumption decision-making. Furthermore, Individual cognition and attitude are also important factors affecting rural residents' consumption decision-making, while values, beliefs, and cultural factors have less obvious effects on consumer decision-making.
There is a contradiction about the impact of cultural factors.
Reply: Thank you for your suggestions. You mentioned that there are conflicting views in the summary, particularly regarding the influence of cultural factors. We carefully reviewed the paper to ensure that the influence of cultural factors on consumption decisions was clearly described and more uniform and clear explanation was provided in the paper. It also emphasizes the differences with urban consumers.
Also, the introduction is hard to follow: The author’s use phrases such as External factors, Internal factors, Phycological factors, Culture, Socio-economic status, Family income and education, etc. These phrases are used in great deal, in repeated fashion throughout the introduction, and seem to be very central to this research. However, they are used so disjointedly that I am unable to get a clear understanding of what motivates this research. Simply put, the ideas don’t flow naturally. I suggest that the author rewrites the introduction in a clear and concise language. In doing so, the authors should highlight, in a less confusing way, the so-call psychological factors and economic factors that are driving the consumption decisions in rural households. The authors can then explain how that is different for urban consumers.
Reply: Thank you for your comments. We have rewritten the introduction to highlight, in clearer and more concise language, the role of psychological and economic factors in driving rural residents' consumption decisions, and to highlight the differences with urban consumers.
Literature Review
- The authors cite some literature that focus on household consumption decisions in the rural areas. Some information on how that contrasts with urban consumers would be interesting to see.
Reply: Thank you for pointing out our problems in time. You mentioned that we cited some literature on rural household consumption decisions, and we would add information on how these rural consumers compare to urban consumers to present this research more fully.
Section 2:
- Section 2 is titled “Psychological Factors of Consumer Decision-Making Among Rural Residents”. However, there is a sub-heading (2.1) on phycological factorsand another sub-heading (2.2) on household income and education level. Is it the author’s proposition that household income and education level are part of psychological factors? If so, my concerns about the language in the introduction is even larger. If not, the authors should rewrite the heading in section 2 to reflect both factors.
Reply: Thank you very much for your comments on our article. In response to your comments, we have changed the title of the original second part to "2. Influencing factors of rural residents' consumption decision-making.”
Research Methodology
Consider using bullet points, roman numerals, or alphabets for the list of steps under “Research Design”. The current use of numbers makes it confusing to follow.
Reply: Thank you for your suggestions. According to your suggestion, I have changed the original numbering form "â‘ , â‘¡, â‘¢..." Change to "a,b,c..." Thus, the experimental situation can be expressed more clearly and intuitively.
Analysis
The analytical section (section 4) is much clearer. The discussion is easier to follow, particularly with the help of the Tables and Figures. Also, Tables 2-4 make clear distinction between internal and external factors influencing consumption decisions. Fig. 2 and Fig. 4 also put more light on the so-called psychological factors. I would like to see this clarity reflected in the introductory part of the paper.
Reply: Thank you for your questions. According to your suggestions, we have regrouped and classified the factors that affect the decision-making of rural residents in the article and explained them in a clearer way in the introduction.
Reviewer 3 Report
Comments and Suggestions for Authors
First of all I would like to congratulate the authors for their article on influencing the consumption decisions of rural residents. It is an article that undoubtedly provides new insights by using a case study. In the interest of improving the quality of the article, I would like to make a few brief recommendations to the authors:
- It would be interesting to indicate whether of the 300 surveys carried out, all of them have been validated or whether there were outliers that had to be eliminated from the final observations. What procedures were used to verify that there were no outliers?
- Has Cronbach's Alpha analysis been carried out for all questions together, including the control variables? If so, this could lead to errors in the estimation of this coefficient.
- What was the consistency of the variables used in the factors such as personal perception, attitude, cultural factors, etc.? It would be interesting to know this information as well.
- Finally, in the discussion section, it would be interesting for the authors to relate the results obtained with the literature initially consulted.
Finally, I encourage the authors to take these reviews into account and to continue with a more robust model to analyse the results obtained in the questionnaire and obtain the corresponding conclusions.
Author Response
Reviewer 3
Comments and Suggestions for Authors
First of all I would like to congratulate the authors for their article on influencing the consumption decisions of rural residents. It is an article that undoubtedly provides new insights by using a case study. In the interest of improving the quality of the article, I would like to make a few brief recommendations to the authors:
Reply: Thank you very much for checking our manuscript and putting forward comments and suggestions.
- It would be interesting to indicate whether of the 300 surveys carried out, all of them have been validated or whether there were outliers that had to be eliminated from the final observations. What procedures were used to verify that there were no outliers?
Reply: Thank you for your comments. Based on your suggestions, our d. Data Analysis in section 3.1 Research Content of the article explains how the study performed outlier clearing. The research uses the outlier query and screening function of SPSS program to analyze and check outliers.
- Has Cronbach's Alpha analysis been carried out for all questions together, including the control variables? If so, this could lead to errors in the estimation of this coefficient.
Reply: Thank you very much for your suggestion. Thank you for your suggestion. According to your suggestion, we corrected the Analysis range of Cronbach's Alpha in 3.2 Analysis of Reliability and Validity of the study
- What was the consistency of the variables used in the factors such as personal perception, attitude, cultural factors, etc.? It would be interesting to know this information as well.
Reply: Thank you for pointing out the problems in the articleWe have added the consistency analysis of the variables used in personal cognition, attitude, cultural factors and other factors to the consistency analysis section of the study.
- Finally, in the discussion section, it would be interesting for the authors to relate the results obtained with the literature initially consulted.
Reply: Thank you for your questions. We have combined the existing results with the literature in the discussion section. A comparison between the findings of the study and the initial literature review was added to the last paragraph of the discussion.
Finally, I encourage the authors to take these reviews into account and to continue with a more robust model to analyse the results obtained in the questionnaire and obtain the corresponding conclusions.
Reply: Thank you very much for your comments and suggestions. We have fully considered your suggestions and made corresponding corrections according to your suggestions. At the same time, in the research model analysis stage, the existing regression analysis was optimized, and the variables were reclassified to make the results more supportive of the research.
Reviewer 4 Report
Comments and Suggestions for Authors
It is a meaningful study to explore the influencing factors of rural residents' consumption decisions. The data source of the paper is reliable, and the structure of the paper is generally acceptable. However, the innovative embodiment of the paper is insufficient, and in addition, there are some shortcomings, which need to be further revised. The specific recommendations are as follows:
1. The author mentions that "education level, household income, consumption attitudes, and sociocultural factors are significant factors influencing rural residents' consumption decisions" (p389), and later mentions that "values, beliefs, and cultural factors have no significant impact on consumption decisions" (p395). What is the difference between socio-cultural factors and cultural factors?
2. This study first analyzed the psychological factors influencing rural residents' consumption decisions in the context of urbanization from six aspects: personal 246 cognition, attitude, values, beliefs, cultural factors, and social influence (P245-248). In this study, 265 respondents were statistically analyzed from the aspects of consumption concept, consumption behavior, household income, and education level, and their consumption patterns and attitudes were discussed (P266). However, in the later analysis, the authors mention that in Figure 5, educational attainment, household income, consumer attitudes, and a combination of 300 factors significantly influence rural residents' consumption decisions. (p300), the author does not analyze his "consumption patterns and attitudes" as described in p266, but only whether each factor influences consumption decisions.
3. In order to further explore the specific psychological factors influencing rural residents' consumption decisions, this study conducted a more in-depth analysis of the overall factors affecting rural residents' consumption decisions, i.e., regression analysis, aiming to investigate the impact of each factor on consumption decisions. (P284)
4. The authors mention that personal cognition and attitudes have a significant positive impact on the consumption decisions of rural residents. However, the influence of values, beliefs, and cultural factors on consumer decisions is not obvious. Social influence also has a positive impact on consumer decision-making (P292). In addition, in "Figure 4 Analysis of the Influence of Psychological Factors on Rural Residents' Consumption Decisions", it is necessary to pay attention to: Are cultural factors and social influence psychological factors? Authors need to carefully consider whether this classification is appropriate.
5. In "Figure 5: Analysis of the impact of family status", the "composite factor" is mentioned, and the previous individual factors are expressed as influential. Note that it would be more appropriate to separate the individual and composite factors.
6. In Figure 5, are the "Consumption Attitudes" the same as the "Consumer Attitudes" in Table 3? In fact, there is a difference between the two.
7. There is redundancy in the article. For example, in the discussion, the author mentions that "however, the influence of values, beliefs and cultural factors on consumer decision-making is not obvious, and social influence also has a certain positive impact on consumer decision-making" (p334), and later mentions that "in addition, the survey data also show that the influence of values, beliefs, and cultural factors on the consumption decisions of rural residents is not very significant". (P342) and so on, there are many places in the article, please revise and improve the author.
8. The first paragraph of the discussion is merely a repetition of the results of the previous analysis. In addition to the specific suggestions for consumer guidance mentioned by the authors, it is suggested that the authors reorganize the discussion and highlight what innovations are in the research in this paper. What are the main academic contributions? How is it different from previous research? What else needs to be improved in the research of this article, etc. Note that in this last section, the author has placed it in the last paragraph of the conclusion, and it is suggested that it would be more appropriate to place it in the discussion. In addition, the discussion of the results of the previous analysis in this paper can be used to discuss the underlying reasons for this.
9. This paper discusses the factors influencing the consumption decision of rural residents from the perspective of consumer psychological factors and family factors. In the conclusion, the authors mention that "the conclusions of this study show that education level, household income, consumption attitudes and sociocultural factors are significant factors influencing rural residents' consumption decisions, while individual cognition and attitudes also have a significant impact." "In fact, is it not well reflected that family factors and personal psychological factors are more important? Or is it just that both are significant? In addition, in the analysis above, the personal psychological factors come first, and the family factors come last, and the order of the conclusions is just reversed. It is recommended that the author adjust the order to achieve logical consistency.
Author Response
Reviewer 4
Comments and Suggestions for Authors
It is a meaningful study to explore the influencing factors of rural residents' consumption decisions. The data source of the paper is reliable, and the structure of the paper is generally acceptable. However, the innovative embodiment of the paper is insufficient, and in addition, there are some shortcomings, which need to be further revised. The specific recommendations are as follows:
Reply: Thank you for your comments and suggestions.
- The author mentions that "education level, household income, consumption attitudes, and sociocultural factors are significant factors influencing rural residents' consumption decisions" (p389), and later mentions that "values, beliefs, and cultural factors have no significant impact on consumption decisions" (p395). What is the difference between socio-cultural factors and cultural factors?
Reply: Thank you for your comments. We have corrected the original "socio-cultural factors" and "cultural factors" in the conclusion to "social factors" and "cultural factors" to further highlight the difference between the two.
- This study first analyzed the psychological factors influencing rural residents' consumption decisions in the context of urbanization from six aspects: personal 246 cognition, attitude, values, beliefs, cultural factors, and social influence (P245-248). In this study, 265 respondents were statistically analyzed from the aspects of consumption concept, consumption behavior, household income, and education level, and their consumption patterns and attitudes were discussed (P266). However, in the later analysis, the authors mention that in Figure 5, educational attainment, household income, consumer attitudes, and a combination of 300 factors significantly influence rural residents' consumption decisions. (p300), the author does not analyze his "consumption patterns and attitudes" as described in p266, but only whether each factor influences consumption decisions.
Reply: Thank you very much for your comments and suggestions. Based on your suggestions, we have added an analysis of whether each factor influences consumption decisions after Figure 5. The marketing effect of each variable on rural residents' consumption decision is further explained.
- In order to further explore the specific psychological factors influencing rural residents' consumption decisions, this study conducted a more in-depth analysis of the overall factors affecting rural residents' consumption decisions, i.e., regression analysis, aiming to investigate the impact of each factor on consumption decisions. (P284)
Reply: Thank you very much for pointing out the problem. We have changed the beginning of 4.2 Regression Analysis to "In order to further explore the specific factors affecting rural residents' consumption decisions, this study conducted a more in-depth analysis of the above overall factors, that is, regression analysis, in order to investigate the influence of each factor." And added the corresponding table of regression analysis results: Table 2-Table 3, so as to more clearly reflect the regression analysis.
- The authors mention that personal cognition and attitudes have a significant positive impact on the consumption decisions of rural residents. However, the influence of values, beliefs, and cultural factors on consumer decisions is not obvious. Social influence also has a positive impact on consumer decision-making (P292). In addition, in "Figure 4 Analysis of the Influence of Psychological Factors on Rural Residents' Consumption Decisions", it is necessary to pay attention to: Are cultural factors and social influence psychological factors? Authors need to carefully consider whether this classification is appropriate.
Reply: Thank you for your suggestions We have changed the cultural factors and social factors of psychological factors to "personal characteristics and learning experience" in the paper, and re-analyzed the updated two factors to better fit the classification of psychological factors.
- In "Figure 5: Analysis of the impact of family status", the "composite factor" is mentioned, and the previous individual factors are expressed as influential. Note that it would be more appropriate to separate the individual and composite factors.
Reply: Thank you for your suggestions. Based on your suggestions, we have separately analyzed the complicating factors in Figure 5. The relationship between each factor is analyzed.
- In Figure 5, are the "Consumption Attitudes" the same as the "Consumer Attitudes" in Table 3? In fact, there is a difference between the two.
Reply: Thank you for your comments. We have corrected the words "Consumption Attitudes" and "Consumer Attitudes" to replace both of them with "Consumption Attitudes".
- There is redundancy in the article. For example, in the discussion, the author mentions that "however, the influence of values, beliefs and cultural factors on consumer decision-making is not obvious, and social influence also has a certain positive impact on consumer decision-making" (p334), and later mentions that "in addition, the survey data also show that the influence of values, beliefs, and cultural factors on the consumption decisions of rural residents is not very significant". (P342) and so on, there are many places in the article, please revise and improve the author.
Reply: Thank you for your suggestions. We have reviewed the full text and deleted or rewritten the superfluous parts. Reduce unnecessary parts of the text.
- The first paragraph of the discussion is merely a repetition of the results of the previous analysis. In addition to the specific suggestions for consumer guidance mentioned by the authors, it is suggested that the authors reorganize the discussion and highlight what innovations are in the research in this paper. What are the main academic contributions? How is it different from previous research? What else needs to be improved in the research of this article, etc. Note that in this last section, the author has placed it in the last paragraph of the conclusion, and it is suggested that it would be more appropriate to place it in the discussion. In addition, the discussion of the results of the previous analysis in this paper can be used to discuss the underlying reasons for this.
Reply: Thank you for your comments. According to your suggestion, we have improved the content of the first paragraph of the discussion section. The innovation of this paper is added, and the contribution and innovation of the research are also explained. The differences with previous studies are also compared. And the shortcomings of the conclusion are deleted.
- This paper discusses the factors influencing the consumption decision of rural residents from the perspective of consumer psychological factors and family factors. In the conclusion, the authors mention that "the conclusions of this study show that education level, household income, consumption attitudes and sociocultural factors are significant factors influencing rural residents' consumption decisions, while individual cognition and attitudes also have a significant impact." "In fact, is it not well reflected that family factors and personal psychological factors are more important? Or is it just that both are significant? In addition, in the analysis above, the personal psychological factors come first, and the family factors come last, and the order of the conclusions is just reversed. It is recommended that the author adjust the order to achieve logical consistency.
Reply: Thank you for pointing out our problems in time. Thank you for your review and valuable comments on this article. Your reminder to the conclusion is very valuable. It is indeed necessary to present the importance of individual psychological factors and family factors in rural residents' consumption decisions in a balanced manner. I reviewed the conclusion when revising the manuscript to more accurately highlight the importance of family factors and to ensure the logic and consistency of the conclusion. In addition, we will emphasize ensuring that the conclusion order is consistent with the logical order of the analysis section to enhance the clarity and readability of the article.
Reviewer 5 Report
Comments and Suggestions for Authors
This paper analysis Influence of Consumption Decisions of Rural Residents in The Context of Rapid Urbanization, the question is interesting, but there still exist some p;aces to be improved.
1. This paper is not correlate with sustainability, it's current form is not in the scope of sustainability;
2.This paper use survey data, author should give out where to get, who were asked, how many people gave quality answer
3.the author should give out the scale of different Classification in Table 1
4.Regression analysis should give the full table of regression
5. In Table 2, only personal cognition and social influence significant level below 10%?
6. The author didn't know which parameter should show to reader in paper
Author Response
Reviewer 5
Comments and Suggestions for Authors
This paper analysis Influence of Consumption Decisions of Rural Residents in The Context of Rapid Urbanization, the question is interesting, but there still exist some p;aces to be improved.
- This paper is not correlate with sustainability, it's current form is not in the scope of sustainability;
Reply: Thank you very much for your comments on our article. We take the sustainability issue you mentioned very seriously. We will consider including more aspects of sustainability in our research to better explore the link between rural consumption decisions and rapid urbanization. And in the article abstract. The discussion and conclusion part integrate the relevant expressions of rural residents' consumption and the stability and sustainability of the party discipline economy.
2.This paper use survey data, author should give out where to get, who were asked, how many people gave quality answer
Reply: Thank you very much for your comments on our article. When we revised the manuscript, we explicitly provided more detailed information about the source of the survey data, the respondents and the number of samples to answer the questions in the third part, so as to increase the transparency and credibility of the research.
3.the author should give out the scale of different Classification in Table 1
Reply: Thank you for your questions. We will supplement the size information for different classifications in Table 1 in the revised manuscript to present the study data more clearly.
4.Regression analysis should give the full table of regression
Reply: Thank you for your questions. We have added a more complete regression analysis table in the fourth part of the paper, so that readers can understand the results more fully.
- In Table 2, only personal cognition and social influence significant level below 10%?
Reply: Thank you for finding the problems in the article in time. We describe the significance levels in Table 4 (formerly Table 2) in more detail and ensure that the findings are accurately reflected.
- The author didn't know which parameter should show to reader in paper
Reply: Thank you for finding the problems in the article in time. We re-examine and ensure that the individual parameters are accurately presented in the paper so that readers can better understand the key content of the research
Round 2
Reviewer 3 Report
Comments and Suggestions for Authors
Congratulations to the authors for the article, and thank them for taking into account the initial comments to improve the quality of the article.
I hope they will continue with the research.
Reviewer 4 Report
Comments and Suggestions for Authors
The authors have made targeted revisions to the paper, and the quality has been significantly improved.
Reviewer 5 Report
Comments and Suggestions for Authors
The author solved all the questions